TECHNICAL RELEASE

# EMImR: a Shiny application for identifying transcriptomic and epigenomic changes

Hiba Ben Aribi[1,*], Careen Naitore[2], Farah Ayadi[1], Souheila Guerbouj[1] and Olaitan I. Awe[3,4,*]

1 Faculty of Sciences of Tunis, University of Tunis El Manar, Tunis, Tunisia
2 Jomo Kenyatta University of Agriculture and Technology, Kenya
3 Department of Computer Science, Faculty of Science, University of Ibadan, Ibadan, Oyo State, Nigeria
4 African Society for Bioinformatics and Computational Biology, Cape Town, South Africa

## ABSTRACT

Identifying differentially expressed genes associated with genetic pathologies is crucial to understanding the biological differences between healthy and diseased states and identifying potential biomarkers and therapeutic targets. However, gene expression profiles are controlled by various mechanisms, including epigenomic changes, such as DNA methylation, histone modifications, and interfering microRNA silencing. We developed a novel Shiny application for transcriptomic and epigenomic change identification and correlation using a combination of Bioconductor and CRAN packages. The developed package, named EMImR, is a user-friendly tool with an easy-to-use graphical user interface to identify differentially expressed genes, differentially methylated genes, and differentially expressed interfering microRNA. In addition, it identifies the correlation between transcriptomic and epigenomic modifications and performs the ontology analysis of genes of interest. The developed tool could be used to study the regulatory effects of epigenetic factors. The application is publicly available in the GitHub repository (https://github.com/omicscodeathon/emimr).

**Subjects** Software and Workflows, Transcriptomics, Ontology and Terminology

**Submitted:** 13 August 2025

\* Corresponding authors. E-mail:
benaribi.hiba@gmail.com;
laitanawe@gmail.com

Preprint submitted at https://doi.org/10.1101/2025.10.16.682862

## INTRODUCTION

Genomics and epigenomics both play significant roles, to a great extent, in all diseases. Indeed, the variations in our DNA and its functions, alone or in combination with the environment that encompasses lifestyle, contribute to disease processes [1, 2].
In this context, studying differentially expressed genes (DEGs) is of utmost importance [3–10].

Gene expression changes are controlled by different mechanisms, including epigenetic modifications that regulate the gene's expression without altering the underlying DNA sequence [11]. Mainly, epigenetic modifications include DNA methylation [12], histone modifications, and microRNA-associated post-transcriptional gene silencing [13]. DNA methylation at the C5 position of cytosine in CpG islands (dinucleotides) is among the central epigenetic mechanisms [14].

In this study, a novel Shiny application, named EMImR, was developed to facilitate the identification and correlation between transcriptomic and epigenomic changes.

**Table 1.** The case study datasets information.

| Dataset | Technique | Samples | Reference |
|---|---|---|---|
| GSE17048 | Expression profiling by array | Whole blood samples from 10 MS patients and 10 healthy donors | [35, 36] |
| GSE106648 | Methylation profiling by array | Whole blood from 10 MS patients and 10 healthy donors | [37] |
| GSE21079 | Non-coding RNA profiling by array | Whole blood from 10 MS patients and 10 healthy donors | [38] |

## METHODOLOGY

### Application development

The shiny application was developed using multiple R (RRID:SCR_001905) packages, including shiny [15], shinydashboard [16], shinythemes [17], shinycssloaders [18], shinyWidgets [19], and shinyFiles [20].

Other R packages were used for data manipulation, including dplyr [21], DT [22], and ggplot2 [23]. The ontology analysis uses the clusterProfiler [24] and enrichplot [25] Bioconductor (RRID:SCR_006442) packages.

The species-specific gene annotation is performed using org.Hs.eg.db for *Homo sapiens* [26], org.Mm.eg.db for *Mus musculus* [27], org.At.tair.db for *Arabidopsis thaliana* [28], org.Dm.eg.db for *Drosophila melanogaster* [29], org.Dr.eg.db for *Danio rerio* [30], org.Rn.eg.db for *Rattus norvegicus* [31], org.Sc.sgd.db for *Saccharomyces cerevisiae* [32], and org.Ce.eg.db for *Caenorhabditis elegans* [33].

### Case study

To validate the pipeline and demonstrate the utility of the developed package, a case study was performed on a publicly available dataset on the GEO database [34]. All these datasets correspond to sequencing data of human blood cell samples, and include healthy individuals and multiple sclerosis (MS) patients. Table 1 includes the information on the analyzed datasets.

MS is an autoimmune demyelinating disease that affects the brain and the spinal cord [39]. It is a multifactorial, neurodegenerative, and inflammatory demyelination disease with incomplete remyelination in the central nervous system. Molecular mechanisms involving epigenetic changes play a pivotal role in the development of MS and influence its progress and susceptibility [40, 41]. Thus, investigation of epigenetic factors of MS can provide new insights into the molecular basis of this disease, which shows a complicated pathogenesis.

RNA sequencing data (Cel file format) was normalized using the limma package [42], annotated, and analyzed with the DESeq2 package [43].

The methylation data (idat file format) was processed using the Minfi R [44] package and annotated. The Limma R [42] package was used to identify differentially methylated CpGs. The final data was filtered based on the location of CpG regions to target promoter-related regions.

The interfering microRNA expression data was analyzed using the GEO2R tool [45].

## RESULTS

### EMImR tool

The tool is publicly available as a shiny application on the project's GitHub repository.



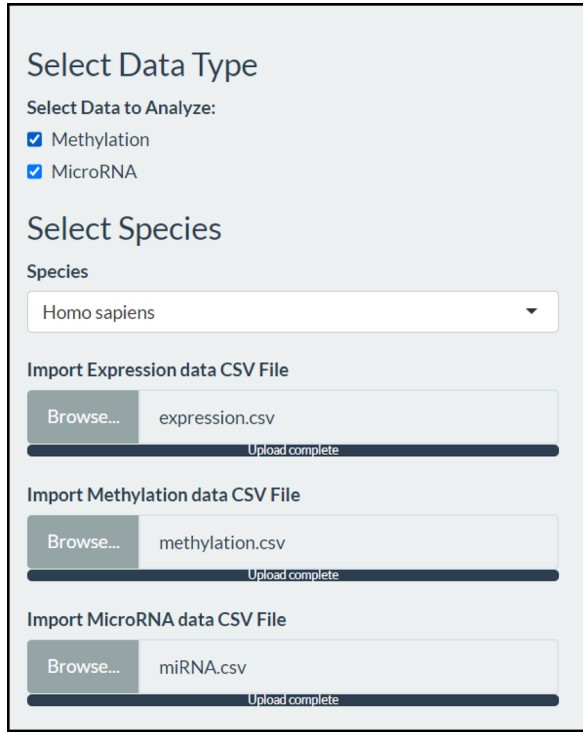

**Figure 1.** The sidebar of the application's user interface (File Import).

The Shiny application is platform-independent, provides an easy-to-use user interface, and does not require any computational skills.

The user interface includes a side panel where the user is requested to define and import the data type to correlate with genetic expression data, as methylation data, microRNA data, or both (Figure 1). Providing gene expression data is mandatory.

The user is requested to select the target species for the ontology analysis. The tool supports eight species, including *Homo sapiens, Mus musculus, Arabidopsis thaliana, Drosophila melanogaster, Danio rerio, Rattus norvegicus, Saccharomyces cerevisiae, and Caenorhabditis elegans.*

The user also needs to define the *p*-value (or *p*-adjust) and the LogFC values to define the DEGs, the differentially methylated genes (DMGs), and the genes associated with differentially expressed microRNAs (DEImRs) (Figure 2).

The outputs are displayed in the application's main panel, which is divided into three sections. In the first section, the differentially expressed genes are visualized in a volcano plot (Figure 3).

In the second section, the DEGs regulated by methylation changes are identified via the intersection of upregulated genes with hypomethylated genes and the intersection of downregulated genes with hypermethylated genes. The results are displayed in a table format (Figure 4).

In the third section, the DEGs regulated by microRNAs are identified by the intersection of upregulated genes with downregulated microRNAs and the intersection of downregulated genes with upregulated microRNAs. The results are also displayed in a table format (Figure 5).

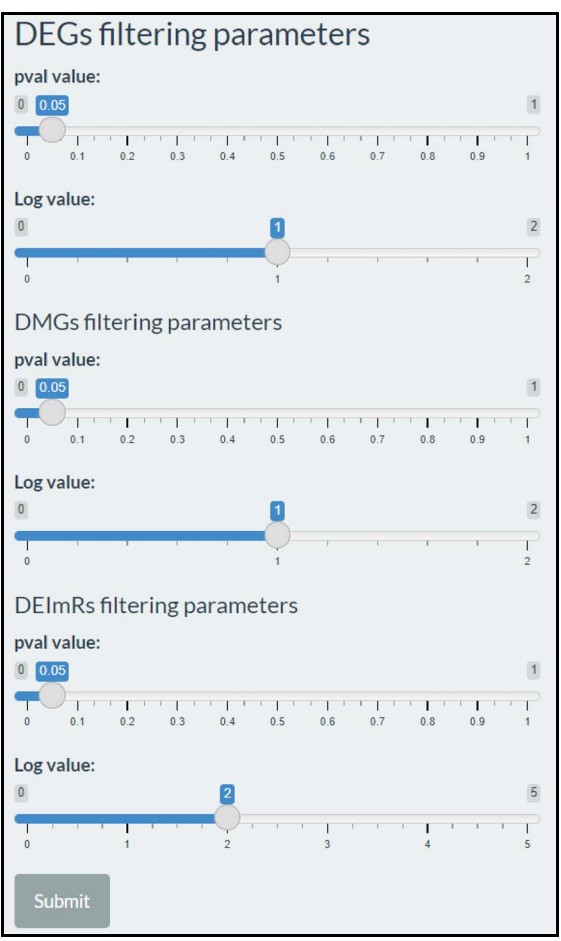

**Figure 2.** The sidebar of the application's user interface (Parameters).

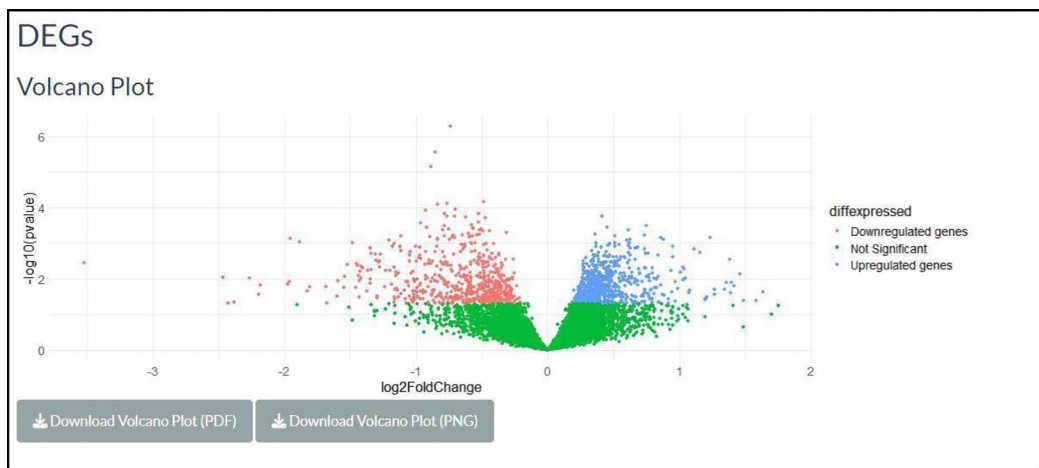

**Figure 3.** Volcano Plot visualizing the DEGs.

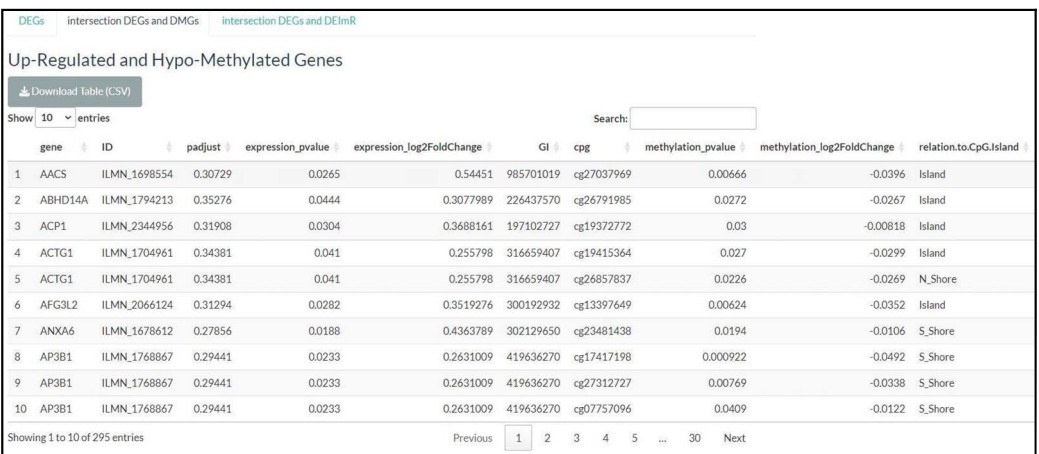

**Figure 4.** Table summarizing the "upregulated and hypomethylated genes" or the "downregulated and hypermethylated genes".

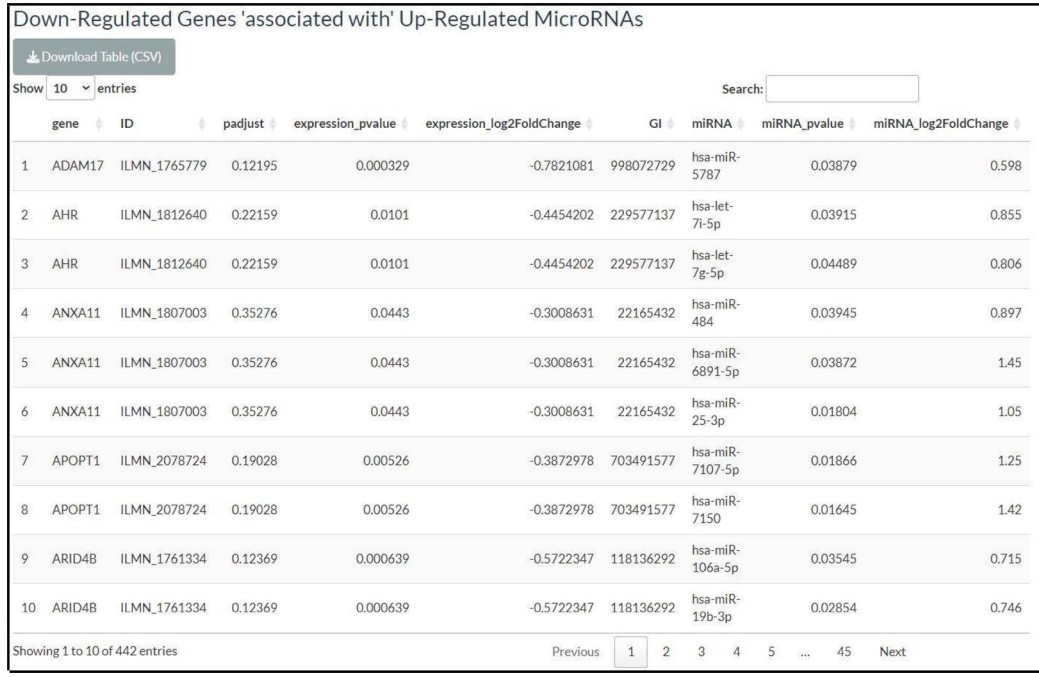

**Figure 5.** Table summarizing the "upregulated genes that are associated with downregulated microRNAs" or the "downregulated genes that are associated with upregulated microRNAs".

The ontology analysis of all four identified gene groups is performed, and the results are visualized in the form of a barplot, dotplot, or cnetplot. The user needs to define the target ontology type from biological processes, cellular components, and molecular functions.

All generated data tables can be downloaded as CSV files, and the generated plots can be downloaded as PDF or PNG files.

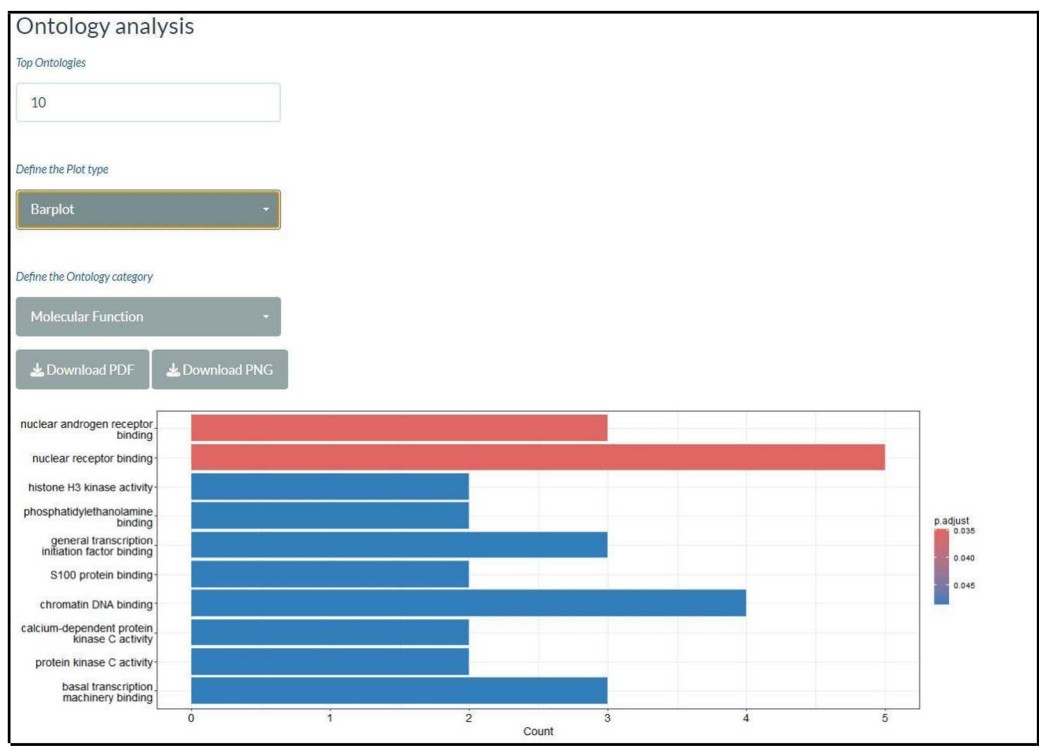

**Figure 6.** Ontology analysis data visualization.

## Case study

The case study aimed to study the transcriptomic and epigenomic changes in MS. Gene expression, methylation, and interfering miRNA data, publicly available datasets on the GEO database, were analyzed using standard analysis pipelines. Then the developed shiny application was used to identify the correlation between transcriptomic and epigenomic modifications. A *p*-value equal to 0.05 and a logFC value of zero were used to filter the DEGs, DMGs, and DEImRs.

A total of 105 genes were both differentially expressed and differentially methylated, and thus potentially regulated by the DNA methylation in the promoter regions of the gene.

Moreover, 27 CpGs in the promoter regions of 26 genes were upregulated and hypomethylated. Four of these genes are implicated in 5 cellular components (including ficoln-1-rich granule lumen, ficoln-1-rich granule, secretory granule lumen, cytoplasmic vesicle lumen, and vesicle lumen). Six of these genes are implicated in the "cytoplasmic translation" biological process, and eight in the molecular process of histone modification. Additionally, 97 CpGs in the promoter regions of 79 genes were downregulated and hypermethylated. Six of these genes play roles in cytoplasmic translation, biological processes, and histone-binding molecular function.

Finally, 37 genes were simultaneously differentially expressed (upregulated) and associated with 37 differentially expressed (downregulated) microRNAs and thus suggested to be regulated by these microRNAs. The genes are implicated in the multiple biological processes represented in the barplot in Figure 6.

This case study highlights the usefulness of the developed tool for analyzing transcriptomic and epigenomic data.

## DISCUSSION AND CONCLUSION

EMImR is a novel tool for the identification of genetic and epigenetic modifications. One of its key strengths is its easy-to-use graphical user interface. The tool is platform-independent and does not require any computational skills for operation, making it accessible across various platforms. All necessary dependencies are automatically installed with the tool. The sole third-party tools requirement is RStudio or an online server, with the prerequisite of having R installed.

The tool supports eight species, including *Homo sapiens, Mus musculus, Arabidopsis thaliana, Drosophila melanogaster, Danio rerio, Rattus norvegicus, Saccharomyces cerevisiae,* and *Caenorhabditis elegans.* The tool is publicly available as a shiny application on the project's GitHub repository.

Future releases of EMImR will extend its functionality beyond data visualization to include genomic and epigenomic data analysis. They will additionally integrate new data types, including spatial and single-cell omics data.

## AVAILABILITY OF SOURCE CODE AND REQUIREMENTS

GitHub Repository: https://github.com/omicscodeathon/emimr
Programming language: R version 4.5.1
License: Artistic license 2.0
Any restrictions to use by non-academics: None
RRID: SCR_027327
Resource consumption: The Shiny application was executed on a Lenovo PC equipped with an Intel® Core™ i5-10210U CPU @ 1.60 GHz, 36 GB of RAM, running a 64-bit Windows 10 operating system. During execution, the application consumed approximately 2.5 GB of memory, utilized up to 24% of CPU, and 1% of GPU resources, as observed in Windows Task Manager.
A video demo is available further demonstrating the features of the tool EMImR: a Shiny Application for Identifying Transcriptomic and Epigenomic Changes. Youtube. https://www.youtube.com/watch?v=n7xYWtWkwU4.

## DATA AVAILABILITY

The supporting data is available in Zenodo [46].

## ABBREVIATIONS

DEImRs, differentially expressed microRNAs; DEG, differentially expressed gene; DMG, differentially methylated gene; MS, multiple sclerosis.

## DECLARATIONS

### Ethical approval and consent to participate

Not applicable.

### Consent for publication

Not applicable.

## Competing interests

The authors declare that they have no competing interests.

## Authors' contributions

HBA conceived the original idea, developed the R package, documented the GitHub repository, and wrote the manuscript. CN, FA, and HBA analyzed the data and validated that the pipeline works. SG contributed in writing the initial manuscript. OIA supervised the project, edited and reviewed the final version of the manuscript.

## Funding

The authors declare that no financial support was received for the research, authorship, and/or publication of this article.

## Acknowledgements

The authors thank the Office of Data Science Strategy (ODSS) of the National Institutes of Health (NIH) and the National Center for Biotechnology Information (NCBI) for their immense support before and during the April 2022 Omics codeathon organized in collaboration with the African Society for Bioinformatics and Computational Biology (ASBCB). The authors acknowledge Danny Lumian for manuscript editing assistance.

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
