## [Reviewer Report]

Indicate in the comments box below whether you are happy with the changes made or if the manuscript is unacceptable.Comments on revised manuscriptThe authors have answered my questions and added new content in the Discussion section as suggested.Indicate in the comments box below whether you are happy with the changes made or if the manuscript is unacceptable.Comments on revised manuscriptThe authors have answered my questions and added new content in the Discussion section as suggested.

---

## [Editor Report]

Editor’s AssessmentCoded and written up as part of the African Society for Bioinformatics and Computational Biology (ASBCB) Omicscodeathons, EMImR is a novel Shiny application for transcriptomic and epigenomic change identification and correlation wrapped up using a combination of Bioconductor and CRAN packages. Case studies are on publicly available GEO data corresponding to sequencing data of human blood cell samples of multiple sclerosis patients to demonstrate how the tool works. And a documentation and videos are provided. Peer review and the study highlighting the usefulness of the developed tool for analyzing transcriptomic and epigenomic data.
*This evaluation refers to version 1 of the preprintEditor’s AssessmentCoded and written up as part of the African Society for Bioinformatics and Computational Biology (ASBCB) Omicscodeathons, EMImR is a novel Shiny application for transcriptomic and epigenomic change identification and correlation wrapped up using a combination of Bioconductor and CRAN packages. Case studies are on publicly available GEO data corresponding to sequencing data of human blood cell samples of multiple sclerosis patients to demonstrate how the tool works. And a documentation and videos are provided. Peer review and the study highlighting the usefulness of the developed tool for analyzing transcriptomic and epigenomic data.
*This evaluation refers to version 1 of the preprint

---

## [Reviewer Report]

Reviewer name and names of any other individual's who aided in reviewerHaikuo LiDo you understand and agree to our policy of having open and named reviews, and having your review included with the published manuscript. (If no, please inform the editor that you cannot review this manuscript.)YesIs the language of sufficient quality?YesPlease add additional comments on language quality to clarify if neededIs there a clear statement of need explaining what problems the software is designed to solve and who the target audience is? NoAdditional CommentsShould be made more clearIs the source code available, and has an appropriate Open Source Initiative license <a href="https://opensource.org/licenses" target="_blank">(https://opensource.org/licenses)</a> been assigned to the code?YesAdditional CommentsAs Open Source Software are there guidelines on how to contribute, report issues or seek support on the code?YesAdditional CommentsIs the code executable?YesAdditional CommentsIs installation/deployment sufficiently outlined in the paper and documentation, and does it proceed as outlined?YesAdditional CommentsIs the documentation provided clear and user friendly?YesAdditional CommentsIs there enough clear information in the documentation to install, run and test this tool, including information on where to seek help if required?Additional CommentsIs there a clearly-stated list of dependencies, and is the core functionality of the software documented to a satisfactory level?YesAdditional CommentsHave any claims of performance been sufficiently tested and compared to other commonly-used packages? Not applicableAdditional CommentsIs test data available, either included with the submission or openly available via cited third party sources (e.g. accession numbers, data DOIs)?Additional CommentsAre there (ideally real world) examples demonstrating use of the software? YesAdditional CommentsIs automated testing used or are there manual steps described so that the functionality of the software can be verified?Additional CommentsAny Additional Overall Comments to the AuthorThe authors developed EMImR as an R toolkit and open-sourced software for analysis of bulk RNA-seq as well as epigenomic sequencing data including DNA methylation seq and non-coding RNA profiling. This work is very interesting and should be of interest to people interested in transcriptomic and epigenomic data analysis but without computational background. I have two major comments: 1. Results presented in this manuscript were only from microarray datasets and are kind of “old” data. Although these data types and sequencing platforms are still very valuable, I don’t think they are widely used as of today, and therefore, it may be less compelling to the audience. It is suggested to validate EMImR using additional more recently published datasets. 2. The authors studied bulk transcriptomic and epigenomic sequencing data. In fact, single-cell and spatially resolved profiling of these modalities are becoming the mainstream of biomedical research since those methods offer much better resolution and biological insights. The authors are encouraged to discuss some key references of this field (for example, PMIDs: 34062119 and 38513647 for single-cell multiomics; PMID: 40119005 for spatial multiomics sequencing), potentially as the future direction of package development.RecommendationMajor Revisions

---

## [Reviewer Report]

Upload additional filesTRR-202508-01R01/stage_files/TRR-202508-01/Review MS/review_gx-TR-1755116555.docxReviewer name and names of any other individual's who aided in reviewerweiming heDo you understand and agree to our policy of having open and named reviews, and having your review included with the published manuscript. (If no, please inform the editor that you cannot review this manuscript.)YesIs the language of sufficient quality?YesPlease add additional comments on language quality to clarify if neededIs there a clear statement of need explaining what problems the software is designed to solve and who the target audience is? YesAdditional CommentsIs the source code available, and has an appropriate Open Source Initiative license <a href="https://opensource.org/licenses" target="_blank">(https://opensource.org/licenses)</a> been assigned to the code?YesAdditional CommentsAs Open Source Software are there guidelines on how to contribute, report issues or seek support on the code?YesAdditional CommentsIs the code executable?YesAdditional CommentsIs installation/deployment sufficiently outlined in the paper and documentation, and does it proceed as outlined?YesAdditional CommentsIs the documentation provided clear and user friendly?NoAdditional CommentsThe Table of Contents on the GitHub page provides a Demonstration Video. However, due to restricted access to YouTube in some regions, it is recommended to also upload a manual in PDF format named “EMImR_manual.pdf” on GitHub.Is there enough clear information in the documentation to install, run and test this tool, including information on where to seek help if required?YesAdditional CommentsIs there a clearly-stated list of dependencies, and is the core functionality of the software documented to a satisfactory level?YesAdditional CommentsHave any claims of performance been sufficiently tested and compared to other commonly-used packages? YesAdditional CommentsIs test data available, either included with the submission or openly available via cited third party sources (e.g. accession numbers, data DOIs)?YesAdditional CommentsAre there (ideally real world) examples demonstrating use of the software? YesAdditional CommentsIs automated testing used or are there manual steps described so that the functionality of the software can be verified?YesAdditional CommentsAny Additional Overall Comments to the AuthorDear Editor-in-Chief, The EMImR developed by the author is a Shiny application designed for the identification of transcriptomic and epigenomic changes and data association. This program is mainly targeted at Windows UI users who do not possess extensive computational skills. Its core function is to identify the intersections between genetic and epigenetic modifications Review Recommendation I recommend that after making appropriate revisions to the current “Minor Revision”, the article can be accepted. However, the author needs to address the following issues. Major Issue The article does not provide specific information on the resource consumption (memory and time) of the program. This is crucial for new users. Although we assume that the resource consumption is minimal, users need to know the machine configuration required to run the program. Therefore, I suggest adding two columns for “Time” and “Memory” in Table 1. Minor Issues 1. GitHub Page The Table of Contents on the GitHub page provides a Demonstration Video. However, due to restricted access to YouTube in some regions, it is recommended to also upload a manual in PDF format named “EMImR_manual.pdf” on GitHub. In step 4 of the Installation Guide, it states that “All dependencies will be installed automaticly”. It is advisable to add a step: if the installation fails, prompt the user about the specific error location and guide the user to install the dependent packages manually first to ensure successful installation. Currently, the command “source(‘Dependencies_emimr.R’)” does not return any error messages, which is extremely inconvenient for novice users. The author can provide the maintainer's email address so that users can seek timely solutions when encountering problems 2. R Version The author recommends using R - 4.2.1 (2022), which was released three years ago. The current latest version is R 4.5.1. It is suggested that the author test the program with the latest version to ensure its adaptability to future developments. 3. Flowchart Suggestion It is recommended to add a flowchart to illustrate the sequential relationships among packages such as DESeq2 for differential analysis, clusterProfiler for clustering, enrichplot for plotting, and miRNA - related packages (this is optional). 4.Function Addition Currently, the program seems to lack a button for saving PDFs, as well as functions for batch uploading, saving sessions, and one - click exporting of PDF/PNG files. It is recommended to add the “shinysaver” and “downloadHandler” functions to fulfill these requirements. 5. Personalized Features and Upgrade Plan To attract more users, more personalized features should be added. The author can mention the future upgrade plan in the discussion section. For example, currently, DESeq2 is used for differential analysis, and in future upgrades, more methods such as PossionDis, NOIseq, and EBseq could be provided for users to choose from. 6. Text Polishing Suggestions 6.1 Unify the usage of “down - regulated” and “downregulated”, preferably using the latter. 6.2 “R - studio version” ---》 “RStudio” 6.3 Lumian, ---》 Lumian 6.4 no login wall ---》 does not require user registration 6.5 Rewrite “genes were simultaneously differentially expressed and methylated” as “genes that were both differentially expressed and differentially methylated”. 6.6 Ensure that Latin names of species are in italics 6.7 make corresponding modifications to other sentences to improve the accuracy and professionalism of the language in the article. The above are my detailed review comments on this article. I hope they can provide a reference for your decision - making. Best regards, weiming he 2025-09-12RecommendationMinor Revisions